# Lateralization of CA1 assemblies in the absence of CA3 input

Hefei Guan [1,2], Steven J. Middleton [1], Takafumi Inoue [2] & Thomas J. McHugh [1✉]

In the hippocampal circuit CA3 input plays a critical role in the organization of CA1 population activity, both during learning and sleep. While integrated spatial representations have been observed across the two hemispheres of CA1, these regions lack direct connectivity and thus the circuitry responsible remains largely unexplored. Here we investigate the role of CA3 in organizing bilateral CA1 activity by blocking synaptic transmission at CA3 terminals through the inducible transgenic expression of tetanus toxin. Although the properties of single place cells in CA1 were comparable bilaterally, we find a decrease of ripple synchronization between left and right CA1 after silencing CA3. Further, during both exploration and rest, CA1 neuronal ensemble activity is less coordinated across hemispheres. This included degradation of the replay of previously explored spatial paths in CA1 during rest, consistent with the idea that CA3 bilateral projections integrate activity between left and right hemispheres and orchestrate bilateral hippocampal coding.

[1] Laboratory for Circuit and Behavioral Physiology, RIKEN Center for Brain Science, 2-1 HirosawaWako-shi Saitama, Japan. [2] Department of Life Science and Medical Bioscience, School of Advanced Science and Engineering, Waseda University, Tokyo 162-8480, Japan. ✉email: thomas.mchugh@riken.jp

The spatial code observed in the rodent hippocampus[1] has been a crucial tool in the dissection of fundamental memory mechanisms[2–4]. Output from the CA3 region of the hippocampus is essential for both rapid memory acquisition in new environments and the recall of an already formed memory in an associative manner[5]. Further, CA3 transmission has been shown to be critical for the normal generation of sharp-wave ripples (SWRs), the coordinated reactivation of cell pairs during SWRs, as well as contextual memory consolidation[6]. Moreover, chronically silencing CA3 output abolishes temporal coding at the ensemble level in CA1, while sparing both rate and temporal coding at the single-cell level[7]. However, it remains unclear whether the experience-dependent sequential "replay" of place cells that occur during SWRs, population events correlated with memory consolidation[8,9], remains intact in the complete and chronic absence of CA3 transmission.

A further gap in our understanding of circuit function is if the functional lateralization observed in the human hippocampus[10,11] is present in rodents and what circuitry may underlie it. Previous work in mice, largely focused on hemispheric differences in receptor expression[12], dendritic morphology, in vitro electrophysiological, and behavior, has argued for lateralization[13], primarily attributed to differences in the bilateral CA3 projections to CA1[14–16]. Further, several in vivo physiological studies, again primarily focused on CA3, have observed some hemispheric differences[17–19]. However, bilateral recordings of place-cell activity in the rat hippocampus consistently report similar and integrated activity in the left and right CA1[4,20]. Given the lack of robust direct contralateral connections in CA1, a parsimonious interpretation of these data is that bilaterally projecting CA3 inputs may be crucial for integrating CA1 activity between the hemispheres; however this has not been directly examined. In this work, we investigated the role of bilateral CA3 input in the lateralization and coordination of bilateral CA1 spatial coding employing the CA3-TeTX transgenic mouse model. Although the properties of single place cells and SWRs in CA1 were comparable bilaterally, we observed a decrease of ripple synchronization between the left and right CA1 after silencing CA3. Further, although cell assemblies, SWRs, and replay all endure following the chronic silencing of CA3 transmission, the properties of ensemble events were significantly altered, not only becoming poorer in quality but also more lateralized. These results suggest the CA3 output is important for the bilateral coordination of downstream CA1 spatial coding during spatial navigation and consolidation.

## Results

Here we investigated the role of CA3 in organizing bilateral ensemble CA1 activity by genetically blocking synaptic transmission at CA3 terminals through the inducible transgenic expression of tetanus toxin (TeTX) and recording from bilateral CA1 in CA3-TeTX mice[5] and littermate controls (CTRs), as mice explored a familiar linear track, as well as during flanking periods of rest (Fig. 1a). Consistent with previous reports[6], we observed that the mean firing rate (FR) of CA1 pyramidal cells (PCs) (Fig. 1b, c) was similar in mutant (MUT) and CTR mice during exploration (CTR $0.40 \pm 0.027$ Hz; MUT, $0.46 \pm 0.028$ Hz) and during rest, both overall (CTR, $0.623 \pm 0.053$ Hz; MUT, $0.621 \pm 0.038$ Hz) and specifically during the occurrence of SWRs (CTR, $1.61 \pm 0.171$ Hz; MUT, $1.46 \pm 0.095$ Hz; Fig. 1c). However, the spatial information (SI) of CA1 place cells during exploration was significantly lower in the MUT mice compared to CTRs, reflecting the expected decrease in spatial acuity in the absence of CA3 input (CTR, $0.87 \pm 0.036$ bits/spike; MUT, $0.79 \pm 0.038$ bits/ spike; Fig. 1c).

**PC properties remain similar across bilateral CA1 after silencing CA3**. We next compared the basic properties of place cells recorded in the left and right CA1 in CTR mice and in MUTs following the silencing of CA3 input, and found that the mean FR and SI during exploration were similar across hemispheres within both genotypes (FR: CTR $L$, $0.41 \pm 0.036$ Hz; CTR $R$, $0.39 \pm 0.041$ Hz; MUT $L$, $0.48 \pm 0.040$ Hz; MUT $R$, $0.44 \pm 0.038$ Hz; SI: CTR $L$, $0.87 \pm 0.048$ bits/spike; CTR $R$, $0.86 \pm 0.055$ bits/spike; MUT $L$, $0.70 \pm 0.041$ bits/spike; MUT $R$, $0.88 \pm 0.063$ bits/spike). The PC FR during rest, as well as during SWRs, also showed no evidence of lateralization in CTRs or MUTs (FR in rest: CTR $L$, $0.55 \pm 0.059$ Hz; CTR $R$, $0.71 \pm 0.095$ Hz; MUT $L$, $0.66 \pm 0.057$ Hz; MUT $R$, $0.58 \pm 0.050$ Hz; FR in SWRs: CTR $L$, $1.43 \pm 0.198$ Hz; CTR $R$, $1.86 \pm 0.304$ Hz; MUT $L$, $1.63 \pm 0.134$ Hz; MUT $R$, $1.43 \pm 0.149$ Hz; Fig. 1d). Together, these data suggest that even in the absence of CA3 input, basic PC firing properties remain similar across hemispheres.

**Decrease in bilateral CA1 cell assemblies following the silencing of CA3 input**. Given that we observed no hemispheric differences in single-cell spatial coding properties in CA1, we next asked how the loss of CA3 transmission impacted the bilateral co-activation of temporally coordinated cell assemblies. We identified the repeated neuronal co-activation of place-cell assembly patterns from bilateral CA1 during exploration of the familiar track using a combination of principal component and independent component analyses (ICAs), and then tracked the activity of these assemblies over time[21] (Fig. 2a). Assemblies were present in the MUT mice, and although no significant changes were observed in average assembly size between the genotypes (CTR, $3.19 \pm 0.099$ cells; MUT, $3.17 \pm 0.116$ cells; Supplementary Fig. 2a), there was an increased fraction of neurons in the CA3-TeTX mice that did not belong to any assembly (CTR, $0.37 \pm 0.025$; MUT, $0.47 \pm 0.040$; Supplementary Fig. 2b). Moreover, compared to the CTRs, the strength of assembly activation in the MUTs was significantly lower during the exploration session (CTR, $343.50 \pm 72.691$; MUT, $142.41 \pm 21.676$). This decrease persisted during the rest session, where assembly reactivation strength was again significantly lower in MUTs (CTR, $27.27 \pm 1.825$; MUT, $25.82 \pm 2.724$; Fig. 2b).

To examine the bilateral participation of neurons within single assemblies and ask if assembly diversity was altered in the MUTs, we compared the fraction of unilateral (containing only cells from a single hemisphere, dashed line in Fig. 2a) and bilateral assemblies (containing cells from both hemispheres, solid lines in Fig. 2a) across the genotypes. We observed that the proportion of unilateral cell assemblies was significantly lower in CTR mice than in MUTs, suggesting a loss of CA3 input may result in decreased bilateral assembly activity. To exclude the possibility that this results from an imbalance in the number of cells recorded in the left and right CA1, each assembly was reconstituted 1000 times by a random draw of all cells recorded in that session and the proportion of unilateral assemblies was recalculated for each shuffle. Although the fraction of shuffled unilateral assemblies was similar between CTRs and MUTs, only in the MUT animals was the observed fraction of unilateral assemblies significantly above this chance value (Observed CTR, $0.3587 \pm 0.0199$, Observed MUT, $0.5805 \pm 0.0594$; Shuffled CTR, $0.3014 \pm 0.0148$, Shuffled MUT, $0.3552 \pm 0.0299$; Fig. 2c, left panel). This was true across the population and within single animals of both genotypes as well (Fig. 2c, right panel).

**SWRs and associated assembly reactivation are lateralized in CA3-TeTX mice**. Given that CA3 is important for the coordination of bilateral cell assembly activity across CA1, we next sought to identify how this lateralization might influence SWRs,

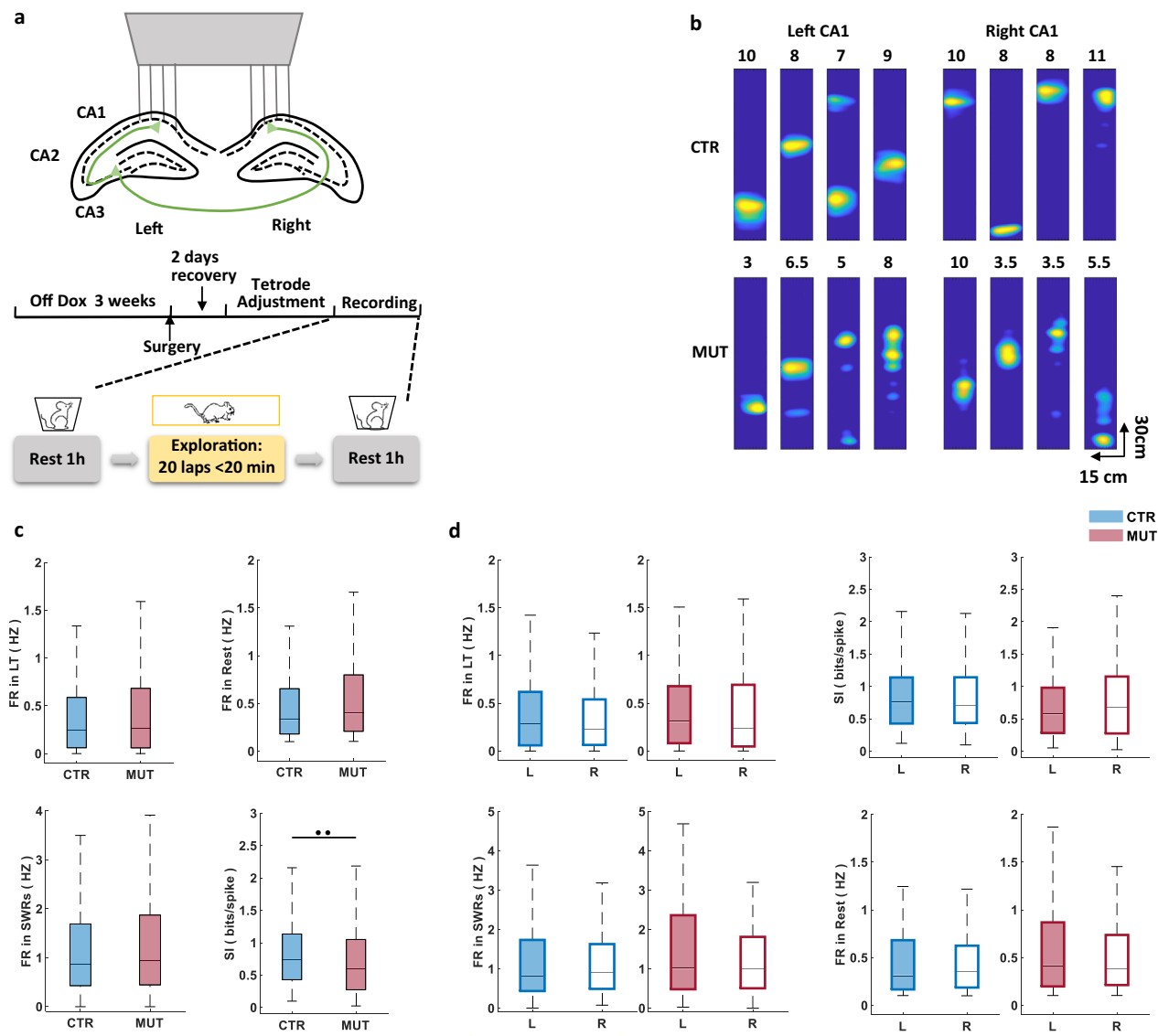

**Fig. 1 Pyramidal cell properties remain similar across bilateral CA1 after silencing CA3. a** Schematic representations of experimental protocols. Top: microdrive placement for bilateral CA1 recording; center: timeline of Dox food withdrawal, surgery, and recording; bottom: structure of the recording day. **b** Representative examples of firing rate maps on the linear track from the left and right CA1 of controls (CTRs) and mutants (MUTs). Colors are scaled to peak firing rate (Hz) indicated at the top of each map (blue, minimum; yellow, maximum). **c** Mean firing rate during exploration (CTR, $N = 4$ animals, $n = 273$ cells; MUT, $N = 5$ animals, $n = 318$ cells; $P$ (mean firing rate during exploration) = 0.1939), rest and within SWRs was comparable between genotypes (CTR, $N = 4$ animals, $n = 227$ cells; MUT, $N = 5$ animals, $n = 290$ cells; $P$ (mean firing rate during rest) = 0.1167; $P$ (mean firing rate during SWRs) = 0.6285). Spatial information during exploration was significantly lower in mutants (CTR, $N = 4$ animals, $n = 273$ cells; MUT, $N = 5$ animals, $n = 318$ cells; $P = 0.0046$). **d** Pyramidal cell properties were comparable between the left and right CA1 (filled bars: left hemisphere; open bars: right hemisphere; CTR, $N = 4$ animals, $n$ (left) = 152 cells, $n$ (right) = 121 cells; MUT, $N = 5$ animals, $n$ (left) = 162 cells, $n$ (right) = 156 cells; mean firing rate during exploration: $P$ (CTR) = 0.6816, $P$ (MUT) = 0.1030; spatial information during exploration: $P$ (CTR) = 0.8559, $P$ (MUT) = 0.5042; during sleep: CTR, $N = 4$ animals, $n$ (left) = 126 cells, $n$ (right) = 101 cells; MUT, $N = 5$ animals, $n$ (left) = 158 cells, $n$ (right) = 132 cells; mean firing rate during rest: $P$ (CTR) = 0.5773, $P$ (MUT) = 0.5291; mean firing rate during ripple: $P$ (CTR) = 0.2369, $P$ (MUT) = 0.0676); significant difference in Fig. 1d were tested with LMMs. Unless mentioned, all significant difference in Fig. 1 were tested with two-sided Wilcoxon's rank-sum test.

which are characterized by the precise temporal organization of neuronal activity, both within and across hemispheres[20]. We first compared the properties of SWRs recorded bilaterally in CA1. Consistent with the previous results in the CA3-TeTX mice[6,7], we observed a decrease in the intrinsic SWRs frequency in the MUT mice, whereas the amplitude, occurrence, and duration were similar to CTRs (Frequency: CTR, 132.9 ± 5.54 Hz; MUT, 108.3 ± 2.01 Hz; Amplitude: CTR, 0.25 ± 0.020 mV; MUT, 0.22 ± 0.026 mV; Duration: CTR, 88.79 ± 5.569 ms; MUT, 81.74 ± 3.152 ms; Occurrence: CTR, 0.21 ± 0.018/s; MUT,

0.17 ± 0.011/s; Supplementary Fig. 3a). Similar to our single-cell data, we found no significant differences in any of these SWR properties between the left and right CA1 in either genotype, although we did note a trend for the right hemisphere SWRs to be slightly larger in amplitude and shorter in duration in the MUT mice (Frequency: CTR $L$, 134.1 ± 6.43 Hz, CTR $R$, 131.8 ± 10.04 Hz; MUT $L$, 105.47 ± 2.136 Hz, MUT $R$, 111.13 ± 3.095 Hz; Amplitude: CTR $L$, 0.22 ± 0.034 mV, CTR $R$, 0.27 ± 0.019 mV; MUT $L$, 0.19 ± 0.012 mV, MUT $R$, 0.26 ± 0.047 mV; Duration: CTR $L$, 87.01 ± 7.022 ms, CTR $R$,

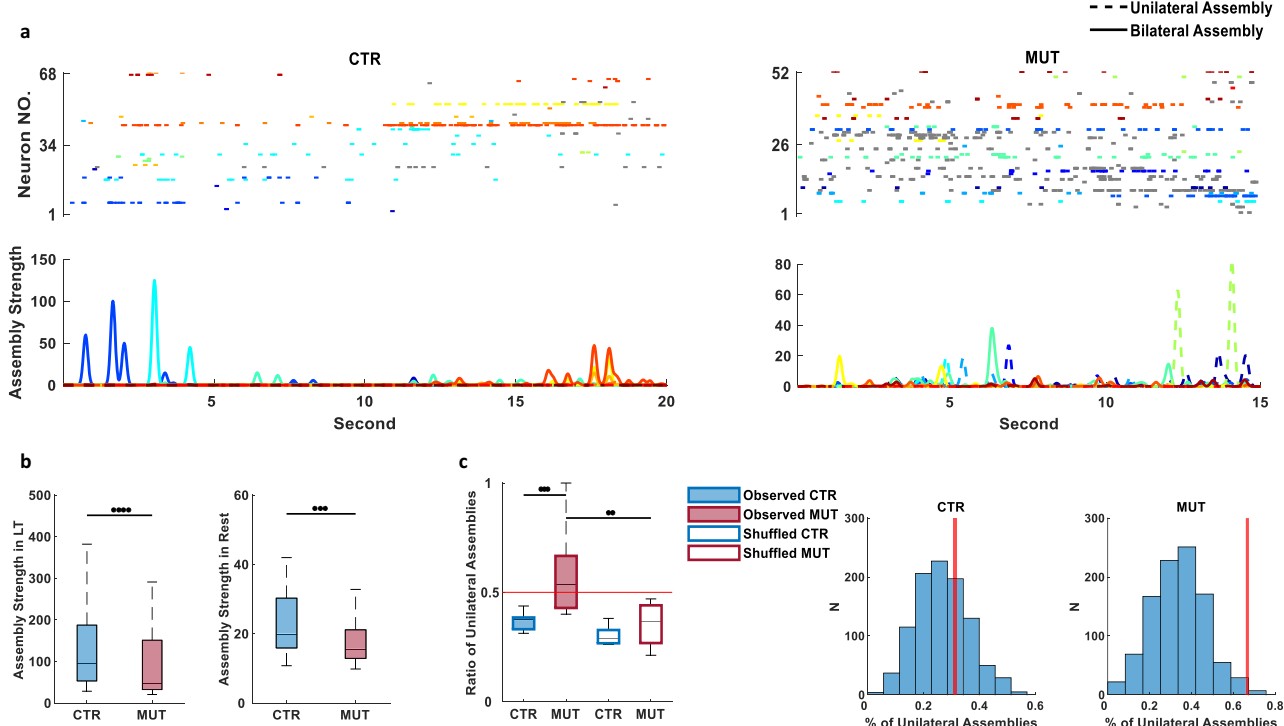

**Fig. 2 Silencing CA3 input decrease bilateral CA1 cell assemblies. a** Top: example raster plots of simultaneously recorded pyramidal cell activity during a single lap on the linear track from one control (68 cells) and one mutant (52 cells) mouse. Cells of the same color belong to the same assembly, cells shown in gray do not participate in any cell assemblies. Bottom: strength of each cell assemblies (colors correspond to rasters above; CTR, 16 assemblies in total; MUT, 12 assemblies in total); solid lines represent assemblies containing cells from bilateral CA1 (CTR, 10 assemblies; MUT, 6 assemblies); dashed line represents unilateral assemblies (CTR, 6 assemblies; MUT, 6 assemblies). **b** Strength of assembly activation on the linear track (LT) and during the subsequent rest session was significantly lower in mutants (In LT: CTR, $N = 4$ animals, $n = 118$ assemblies; MUT, $N = 5$ animals, $n = 119$ assemblies, $P$ (LT) $= 3.196 \times 10^{-7}$; In rest session: CTR, $N = 4$ animals, $n = 114$ assemblies; MUT, $N = 5$ animals, $n = 114$ assemblies, $P$ (rest) $= 1.75 \times 10^{-5}$). **c** Fewer bilateral cell assemblies in CA3-TeTX mice. Left: the fraction of unilateral assemblies was significantly higher in mutants (CTR, $N = 4$ animals, $n = 8$ (two directions for each mouse); MUT, $N = 5$ animals, $n = 10$ (two directions for each mouse); $P = 0.0388$, two-way ANOVA, followed by post hoc Bonferroni test, $P$ (observed CTR vs. observed MUT) $= 0.002$, $P$ (observed MUT vs. shuffled MUT) $= 8.112 \times 10^{-4}$, $P$ (observed CTR vs. shuffled CTR) $= 1$). Right: examples from one control and one mutant animal demonstrate the relationship between the observed fraction of unilateral assemblies (red line) and population distribution of the shuffled reconstituted assemblies. Unless mentioned, all significant difference in Fig. 2 were tested with two-sided Wilcoxon's rank-sum test.

$90.56 \pm 9.660$ ms; MUT $L$, $82.00 \pm 1.056$ ms, MUT $R$, $81.48 \pm 6.600$ ms; Occurrence: CTR $L$, $0.21 \pm 0.030$/s, CTR $R$, $0.21 \pm 0.024$/s; MUT $L$, $0.16 \pm 0.011$/s, MUT $R$, $0.18 \pm 0.018$/s; Supplementary Fig. 3b). More strikingly however, although SWRs in the left and right hemispheres showed strong temporal coordination in CTR mice, the cross-correlation of SWR events in the MUTs demonstrated a dramatic decrease (CTR, $3.38 \pm 0.226$ SWRs/s; MUT, $0.87 \pm 0.063$ SWRs/s; Fig. 3a), suggesting that blockade of CA3 input impaired the synchronization of bilateral CA1 population activity during rest.

We next focused on the activity of cell assemblies specifically during SWRs and found the rate of assembly reactivation was significantly lower in MUTs than CTRs (CTR, $0.085 \pm 0.0038$ Hz; MUT, $0.067 \pm 0.0041$ Hz); however, the overall strength of assembly reactivation was significantly higher (CTR, $118.64 \pm 20.682$; MUT, $243.61 \pm 32.124$; Fig. 3b). We then classified single assemblies as the left or right assemblies, according to which hemisphere contributed the majority of units. We compared the reactivation of left assemblies with that of right assemblies during the right hemisphere SWRs. In CTR mice, reactivation was similar between the left and right assemblies (Strength: $L = 91.10 \pm 8.425$, $R = 178.35 \pm 65.182$; rate: $L = 0.082 \pm 0.0046$ Hz, $R = 0.084 \pm 0.0082$ Hz; Supplementary Fig. 4a); however, in the MUT mice, the reactivation rate was significantly higher during SWRs in the same (right) hemisphere that dominated

the assembly (right assemblies) (Strength: $L = 304.58 \pm 58.467$, $R = 204.00 \pm 41.284$; rate: $L = 0.052 \pm 0.0048$ Hz, $R = 0.074 \pm 0.0069$ Hz; Supplementary Fig. 4b).

**CA3 input is required for CA1 bilateral replay sequences.** Previous research has shown the temporally coordinated SWR-associated replay of place-cell sequences observed during behavior is coordinated across hemispheres[20]. Given that the silencing of CA3 input decreased not only the synchronization of bilateral SWRs but also the co-activation of bilateral CA1 cell assemblies during rest, we asked whether CA3 input is critical for the coordination of bilateral CA1 neurons in replay sequences. We first used the combined activity of CA1 PCs recorded bilaterally during SWR periods to identify significant replay event sequences (Fig. 4a). Despite the loss of CA3 transmission, replay was observed in the MUT mice; however, the quality of these events was significantly worse (Z-score of weighted correlation: CTR, $2.25 \pm 0.028$; MUT, $2.12 \pm 0.026$; Fig. 4b). Further, compared to CTRs, on average the replay events in the MUTs were faster in decoded speed (CTR, $6.64 \pm 0.186$ m/s; MUT, $7.44 \pm 0.213$ m/s; Fig. 4c), covered longer paths (CTR, $78.53 \pm 1.753$ cm; MUT, $83.04 \pm 2.060$ cm; Fig. 4c), and exhibited larger mean jump distances between adjacent decoded positions in MUT mice (CTR, $15.06 \pm 0.353$ cm; MUT, $17.37 \pm 0.438$ cm; Fig. 4c).

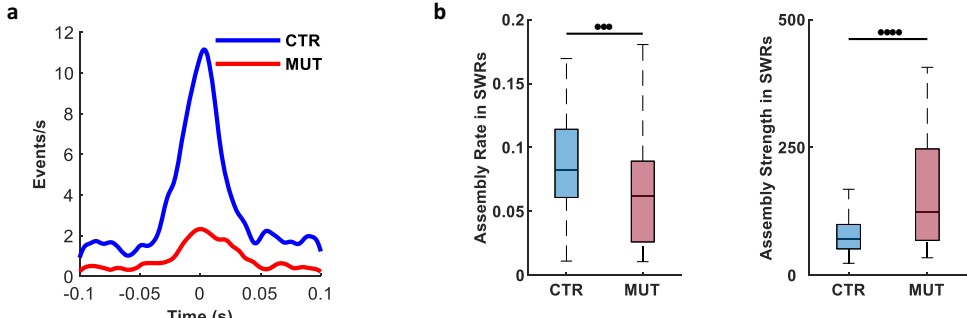

**Fig. 3 CA3 input is required for bilateral coordination of SWRs and assembly reactivation in CA1. a** Cross-correlation of SWRs across hemispheres demonstrates a loss of temporal coordination in CA3-TeTX mice (CTR, $N = 4$ mice, $n = 4$ pairs of tetrodes; MUT, $N = 5$ mice, $n = 5$ pairs of tetrodes; $P = 5.24 \times 10^{-30}$). **b** Assembly reactivation rate during SWRs was significantly reduced in mutants (CTR, $N = 4$ mice, $n = 115$ assemblies; MUT, $N = 5$ mice, $n = 118$ assemblies; $P = 1.27 \times 10^{-7}$); however, strength was significantly increased ($P = 2.81 \times 10^{-4}$). All significant difference in Fig. 3 were tested with two-sided Wilcoxon's rank-sum test.

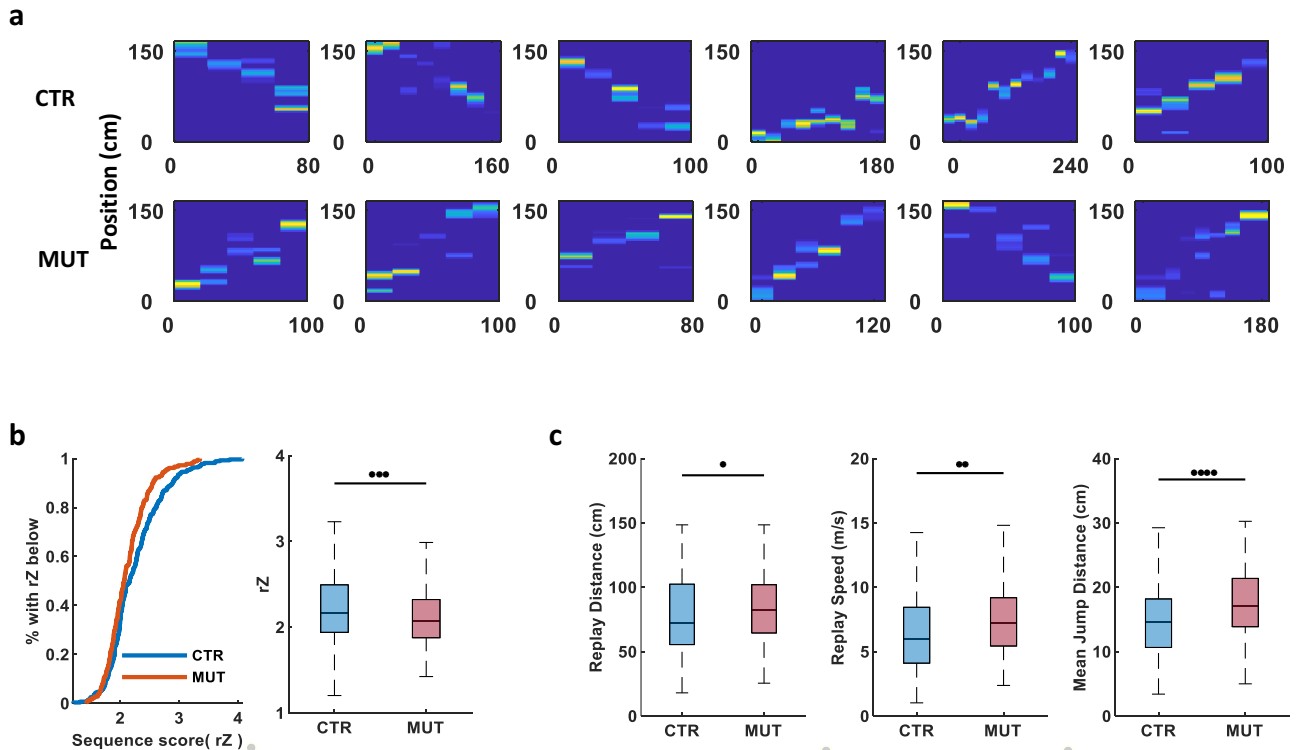

**Fig. 4 Silencing CA3 input alters the replay sequences in bilateral CA1. a** Examples of trajectory sequences derived from Bayesian decoding of spiking activity from bilateral CA1 during SWRs periods. Top: controls; bottom: mutants. Colors are scaled to posterior probability of each significant decoded replay event in 20 ms time bins (blue, minimum; yellow, maximum). **b** The quality of significant replay events decoded with cells from bilateral CA1 during SWRs detected in either hemispheres was reduced in mutant mice (left: population distribution, right: box plot; CTR: $N = 4$ animals, $n$ (right ripple) $= 166$ events, $n$ (left ripple) $= 111$ events, $n$ (combined) $= 277$ events; MUT: $N = 5$ animals, $n$ (right ripple) $= 103$ events, $n$ (left ripple) $= 79$ events, $n$ (combined) $= 182$ events; $P$ (rZ) $= 0.0037$). **c** The distance, speed, and mean jump distance between adjacent positions of significant replay events decoded with cells from bilateral CA1 during SWRs detected in either hemispheres was higher in mutant mice (CTR: $N = 4$ animals, n (right ripple) $= 166$ events, $n$ (left ripple) $= 111$ events, n (combined) $= 277$ events; MUT: $N = 5$ animals, $n$ (right ripple) $= 103$ events, $n$ (left ripple) $= 79$ events, $n$ (combined) $= 182$ events; $P$ (replay speed) $= 0.0012$, $P$ (replay distance) $= 0.0408$, $P$ (mean jump distance) $= 1.1602 \times 10^{-5}$). All significant difference in Fig. 4 were tested with two-sided Wilcoxon's rank-sum test.

Finally, the presence of replay in both genotypes allowed us to compare the contribution of the left and right CA1 cells to each replay event and assess their lateralization. We focused on significant replay events decoded using bilaterally recorded neurons that were detected during SWRs recorded in the same or opposite hemisphere, and determined each place cell's contribution to the sequence by calculating the per cell contribution (PCC) score averaged across all significant

events[22]. In CTR mice, CA1 neurons demonstrated similar contribution to replays occurring during SWRs in the same and opposite hemisphere (CTR same, $4.86 \pm 0.244$; CTR opposite, $5.33 \pm 0.294$); however, in MUT mice, the PCC scores of cells during ripples from the same hemisphere were significantly higher than those from the opposite hemisphere (MUT same, $5.76 \pm 0.335$; MUT opposite, $4.57 \pm 0.272$; Fig. 5a), suggesting that silencing CA3 weakens coordinated memory replay across

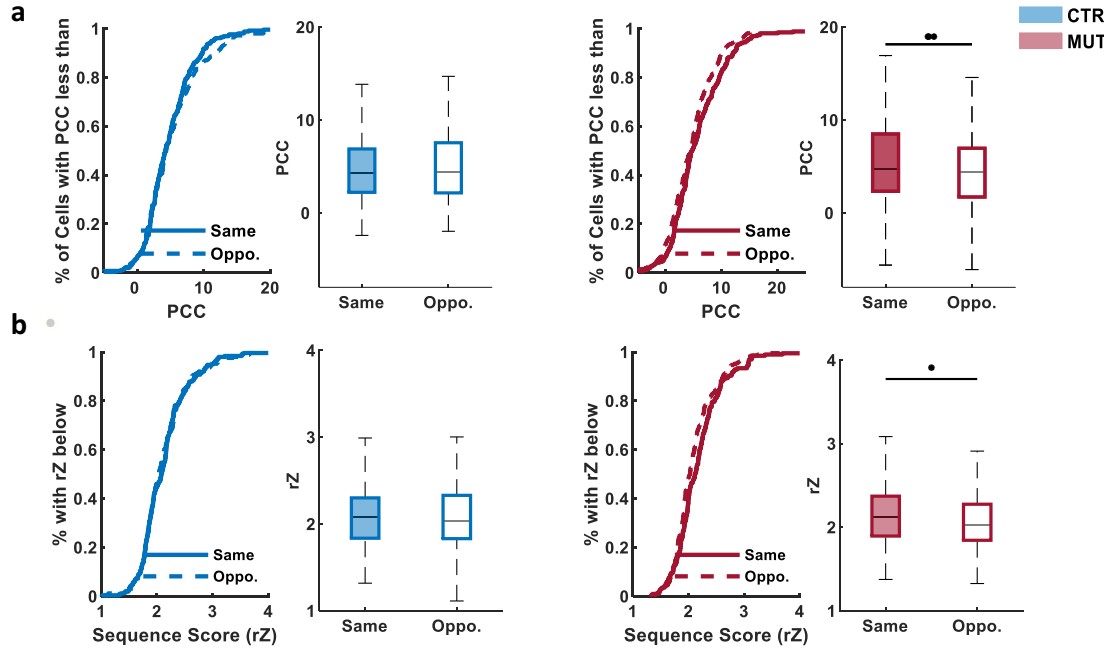

**Fig. 5 CA3 input is required for bilateral coordination of replay sequences in CA1. a** Per cell contribution (PCC) of single neuron to replay events during SWRs from the same and opposite hemisphere was similar in control mice, but in mutants neurons from the same hemisphere with ripple had significantly higher contribution (CTR, $N = 4$ mice, $n$ (left cell + left ripple) = 145, $n$ (right cell + left ripple) = 116, $n$ (left cell + right ripple) = 139, $n$ (right cell + right ripple) = 115, $n$ (same hemisphere) = 260, $n$ (opposite hemispheres) = 255, $P = 0.5167$; MUT, $N = 5$ mice, $n$ (left cell + left ripple) = 155, $n$ (right cell + left ripple) = 148, $n$ (left cell + right ripple) = 118, $n$ (right cell + right ripple) = 117, $n$ (same hemisphere) = 267, $n$ (opposite hemispheres) = 271, $P = 0.0061$). **b** Significant replay events decoded with cells from a single hemisphere during SWRs detected in the same or opposite hemisphere were of similar quality in control mice, whereas in mutants replay quality was significantly lower when only contralateral neurons were used (CTR: $N = 4$ animals, $n$ (left cells + left ripple) = 136 events, $n$ (right cells + right ripple) = 112 events, $n$ (same hemisphere in total) = 248 events; $n$ (left cells + right ripple) = 186 events, $n$ (right cells + left ripple) = 65 events, $n$ (opposite hemisphere in total) = 251 events, $P = 0.8796$; MUT: $N = 5$ animals, $n$ (left cells + left ripple) = 99 events, $n$ (right cells + right ripple) = 116 events, $n$ (same hemisphere in total) = 196 events; $n$ (left cells + right ripple) = 127 events, $n$ (right cells + left ripple) = 80 events, $n$ (opposite hemisphere in total) = 207 events, $P = 0.0346$). All significant difference in Fig. 5 were tested with LMMs.

bilateral CA1. To examine this directly, we decoded place-cell spikes from a single hemisphere during SWRs detected either in the same (ipsilateral) or opposite (contralateral) hemisphere. Once more, in CTR mice, the replay quality was similar in the same and opposite groups; however, in the CA3-TeTX mice, replay quality was significantly lower when only neurons from the opposite hemisphere were used (CTR same, $2.13 \pm 0.028$; CTR opposite, $2.12 \pm 0.028$; MUT same, $2.17 \pm 0.029$; MUT opposite, $2.10 \pm 0.026$; Fig. 5b).

## Discussion

By exploiting the ability to chronically silence CA3 transmission, we demonstrate that CA3 activity plays a key role in the bilateral coordination of CA1 spatial coding, both during behavior and periods of consolidation. We observed that although cell assembly activity, SWRs, and replay all endure following the chronic silencing of CA3, their properties are altered, becoming poorer in quality and more lateralized. Interestingly, the replay trajectories in the MUT mice were faster, longer, and exhibited greater jump distances, consistent with the inability of the CA3 recurrent attractor network to contribute to their structure[23]. Moreover, we observed a loss of temporal coordination between SWR events across the left and right hemispheres, leading to an impairment and lateralization in the re-expression of bilateral sequences of place-cell activity following exploration.

The phenotypes observed in the CA3-TeTX mice are distinct from those seen in experiments using optogenetics to transiently silence the CA3 axon terminals, which resulted in a decrease in the occurrence of SWRs in addition to poorer replay quality

across CA1[24]. In contrast to the fast transient local inhibition provided by optogenetics, our transgenic approach allows us to induce the silencing of CA3 in adult mice just prior to recording in a comparatively slow manner. Thus, there may be compensation due to the chronic nature of the manipulation, which allows SWRs and temporal spike coding, albeit with altered properties, to persist, even in a more lateralized manner. Despite this, it is important to note that our transgenic mice permit silencing of transmission across all CA3 PCs[5], providing a more complete and reproducible model to study these phenomena and answer the question of exactly what CA3 input contributes. We believe that the complementary nature of acute and chronic approaches has established that both are critical for understanding the importance of this circuit and its role in memory.

Although it is widely accepted that CA3 is the primary driver of SWRs and replay events, there is evidence that SWRs can be triggered by other inputs, including those from CA2[25,26], the subiculum[27], and perhaps also intrinsically generated within CA1 itself[28]. Moreover, recent in vivo evidence has established that activity in CA2 contributes to both SWR generation and reactivation of cell assemblies, even when CA3 is intact[25,26]. Single-unit recordings from all CA subregions demonstrate that prior to the characteristic bursting observed in CA3, which classically signaled the onset of SWRs, the CA2 PC population can display a biphasic activity pattern preceding its neighbors[25]. This activity, with brief temporal delays, is then successively evident in areas CA3a, CA3b, CA3c, and finally CA1. Moreover, Oliva et al.[26] also demonstrated structured reactivation of socially active neurons during these CA2 SWR events, as well as evidence that

manipulation of these events could impact memory. Further, a very recent study provided both in vitro and in vivo evidence that the subiculum can serve as a secondary independent SWR generator, leading to events that propagate backwards into CA1. Although that study did not examine replay, it raises the possibility that SWRs seen in the absence of CA3 transmission may also have a subicular origin[27]. Finally, there is evidence from in vitro studies that CA1 can generate SWRs independently, when all other hippocampal subfields are dissected away[28]. With our current data, we cannot distinguish between these possibilities. However, given that CA3 acts primarily to inhibit CA2 in vivo[29] and even under normal conditions CA2 contributes to a proportion of normal SWRs and replay in CA1[30], our belief is that CA2 most likely acts to initiate SWRs in the absence of CA3 output.

Although the lateralization of function within the human hippocampus has been clearly demonstrated[10,11], lateralization studies in rodent hippocampus have provided mixed evidence. Lateralization of anatomical connectivity, spine size, and capacity for synaptic plasticity have been noted in the mouse hippocampus, mainly focusing on the CA3 to CA1 circuits[12,13]. Further, previous in vivo physiological studies focused on hippocampal lateralization have noted differences in both oscillations and function between the hemispheres[17–19], including differential contributions of the left and right CA3 during the retrieval of spatial working memory[18]. Fiber photometry recording of bilateral CA3 activity in mice performing a spatial working memory T-maze task revealed that neurons in the left CA3 showed more activity than in the right CA3 specifically during the choice phase. In addition, performance was impaired only when neurons in the left CA3 were silenced using optogenetics[18]. In another study, γ-oscillations recorded from bilateral CA3 under anesthesia were found to be asymmetric and lateralized, with higher amplitude in the right hemisphere[17]. The synchronization of γ-oscillation from bilateral CA1 was disrupted by blocking the interhemispheric connection between bilateral CA3, suggesting the left and right sides of the hippocampus do not transfer the same information[17]. Villalobos et al.[19] also performed local-field potential recording from bilateral CA1 in rats during slow-wave sleep and reported that ripple oscillations from the left and right hippocampi are asynchronous. Here we observed a much higher degree of L/R ripple temporal synchrony in our CTR mice, consistent with earlier reports in rats[31], suggesting that species, strain, or analytic differences may underlie these divergent results. Despite the lateralization reported on the level of oscillations, the interhemispheric integration of spatial representations during exploration and rest has been repeatedly observed[4,20], suggesting integration of the hippocampal spatial code. Our data supports this conclusion, as we observed no lateralization between the left and right sides of the hippocampus on both the single cell and ensemble level in CTR mice; however, the increase in lateralization in the CA3-TeTX animals suggests that the CA3 to CA1 projections may be responsible for this function. It is important to note that a limitation of the CA3-TeTX model to date is that the kinetics and efficacy of the transgenic blockade of transmission has only been verified in male mice[5]; thus, in this study, female mice were not included. Previous work has noted no sex differences in hippocampal lateralization[32]; however, future work should address the role of CA3 transmission in the integration of the CA1 spatial code in female mice.

Although earlier behavioral characterization of the CA3-TeTX mice identified learning deficits in novel environments, suggesting that CA3 output plays an important role in rapidly forming representation of a novel context[5], as well as in pattern completion-based recall[5], the MUT mice performed normally during training and recall in the Morris water maze[5]. It has been suggested that the left and right CA3 may play different roles during the acquisition and retrieval of a spatial memory, with the left CA3 being more responsible for the storage of discrete locations within an environment and discrimination of distinct places during retrieval, and right CA3 more responsible for the integration of route information in spatial working memory tasks[13]. Although the intact water maze learning in the CA3-TeTX mice, despite their increased lateralization, does not directly support this hypothesis, a caveat of the CA3-TeTX mouse model is that we cannot disambiguate the role of bilateral CA3 to CA1 projections from that of contralateral CA3 to CA3 projections in the lateralization observed here. Thus, while this work highlights the importance of CA3 transmission in the maintenance of neural ensemble coordination between the left and right CA1, future work with more spatially targeted approaches remains necessary.

## Methods

**Subjects**. Five male CA3-TeTX transgenic mice[5] and four male CTR littermates, all aged between 16 weeks and 24 weeks, were used in this study. All mice were bred and genotyped as previously reported[5]; in brief, male homozygous double transgenic mice (KA1-Cre/KA1-Cre, TetO-TeTX/TetO-TeTX) were bred with females heterozygous for the αCamKII-loxP-STOPloxP-tTA/+ transgene. Fifty percent of the the resultant progeny are heterozygous triple transgenic mice (KA1-Cre/+, TetO-TeTX/+, αCamKII-loxP-STOP-loxP-tTA /+), which are referred to as CA3-TeTX mice, whereas the remaining 50% are heterozygous double transgenic mice (KA1-Cre/+, TetO-TeTX/+, +/+), which serve as CTR mice. Genotyping of tail DNA by PCR to detect the presence of each transgene separately was conducted with the following primers: Tg1, 5′-AAATGGTTTCCCGCAGAACC-3′ and 5′-CTAAGTGCCTTCTCTACACC-3′. For Tg2, 5′-CGCTGTGGGGCATTT-TACTTTAG-3′ and 5′-GGGTCCATGGTGATACAAGG-3′. For Tg3-TeTX and Tg3-GFP, 5′-GTGGCGGATCTTGAAGTTCACC-3′ and 5′-GACCCTGAAGTT-CATCTGCACC-3′. Identical PCR conditions (94 °C for 2 min; 94 °C for 5 s, 58 °C for 1 min, 72 °C for 1 min × 35 cycles; 72 °C for 7 min) were used for all primer pairs. All mice were maintained on doxycycline (10 mg/kg) containing diet from conception until 3 weeks before the microdrive implantation surgery. At the time they were switched to normal chow to allow transgenic expression of TeTX in CA3 PCs of the triple transgenics. Mice were maintained in a room with a temperature of 22° and humidity of 48% on a 12 h light–dark cycle, with lights on from 8:00 to 20:00 h. All procedures were approved by the RIKEN Institutional Animal Care and Use Committee. Data collection and analyses were not performed blind to the conditions of the experiments, nor was data collection randomized.

**Surgery and recording preparation**. A custom microdrive was fabricated with two separate bundles, 2.5 cm apart from each other, with each bundle containing eight tetrodes arranged into a linear array and each tetrode independently adjustable prior to recording (Fig. 1a). During surgery, the microdrive was stereotaxically implanted over CA1 bilaterally, with the center of each bundle targeted at −1.94 mm posterior and ±1.25 mm lateral of the bregma. Eight adjustable recording tetrodes were implanted in left CA1 and eight tetrodes in right CA1 during implantation surgery (Supplementary Fig. 1). Following a recovery periods of 2 days after surgery, tetrodes were slowly lowered independently towards the CA1 pyramidal layer. Recording commenced when all tetrodes were located in the PC layer, this was evident by large amplitude PC spikes and spontaneously occurring sharp-wave ripples during immobility.

**Recording protocol**. Data were collected using the Cheetah (Neuralynx) data acquisition software. Three recording sessions were conducted over the course of a single day. Animals were first placed in a highly familiar, small, high-walled enclosure (sleep box) for 1 h to ensure stability of a single-unit recording. Then, animals were placed on a familiar linear track with opaque walls, measuring 150 cm (length) × 15 cm (width) × 15 cm (height). Animals could freely explore the linear track, moving from one end to the other, and recordings continued until either 20 laps had been completed or 20 min had elapsed. Animals were then immediately transferred back to the same sleep box for a period of 1 h. Position and head direction were tracked throughout all three sessions using a pair of red and green light-emitting diodes affixed above the microdrive and monitored by a ceiling mounted camera.

**Place-cell analyses**. SpikeSort3D software (Neuralynx) was used to cluster cells manually based on recorded waveform parameters. To compare the same group of cells across the three sessions, cells in the last rest session were clustered first and those cluster boundaries were applied to the first two sessions. Clusters with > 0.5% of spikes displaying an interspike interval < 2 ms, containing < 75 total spikes or

having a cluster isolation distance < 20[33] were discarded. Of the remaining cells, those with a mean spike width exceeding 200 µs and having a complex spike index[34] over 5, were classified as putative PCs. A velocity threshold of 5 cm/s was then applied to the data on the linear track and any cell with < 75 spikes during periods of movement were also discarded.

**Firing rate**. FR on the linear track was calculated as the total number of spikes of each cell divided by the total time that animal spent in the linear track. FR in rest was calculated as the total number of spikes of each cell divided by the total time that animal spent in rest session. FR in SWRs was calculated as the total number of spikes of each cell during ripples divided by the duration of all ripples in rest session.

**Spatial information**. Spikes of each place cell were separated into two groups according to the direction of the animal's movement. The position at which spikes occurred was determined by binning the track into 100 equally sized bins and the resultant spike map was smoothed with a Gaussian smoothing kernel (SD = 5 cm). A FR map was derived by diving the smoothed spike map by the occupancy map, which was smoothed with the same Gaussian smoothing kernel. SI was calculated as described previously[35]:

$$\text{Spatial information} = \sum_i p_i \frac{\lambda_i}{\lambda} \log_2 \frac{\lambda_i}{\lambda} \tag{1}$$

where $i$ is a single spatial bin, $p_i$ is the probability of the animal being in that bin, $\lambda_i$ is the mean FR in that bin, and $\lambda$ is the mean FR of the neuron.

### Cell assembly analyses

*Assembly pattern identification*. Cell assemblies were detected via the co-activation of place cells, while the mice were exploring the familiar linear track using previously described methods[21]. The spike activity of each neuron was separated into the left and right directions according to the movement direction of animal, temporally binned into 25 ms windows, and the number of spikes in each bin was calculated and normalized with a z-score transform. Principal component analysis was then applied to this z-scored spike time matrix (Z):

$$\sum_{j=1}^{n} \lambda_j p_j p_j^T = \frac{1}{n} Z Z^T \tag{2}$$

where $p_j$ is the $j$th principal component with the corresponding eigenvalue $\lambda_j$ and $\frac{1}{n} zz^T$ is the correlation matrix of $Z$. The Marčenko–Pastur law was used to estimate the number of significant patterns, which means that for an $n \times B$ matrix, the eigenvalues are expected to exceed $\lambda_{\max}$, if the firing activity of the neurons are independent from each other[36]. Here, $\lambda_{\max}$ is defined by

$$\lambda_{\max} = \left(1 + \sqrt{n/B}\right)^2 \tag{3}$$

The number of eigenvalues above $\lambda_{\max}$, which represents the number of distinct significant patterns, was defined as $N_A$. The significant principal components were projected onto the binned spike time matrix (Z):

$$Z_{\text{PROJ}} = P_{\text{SIGN}}^T Z \tag{4}$$

where $P_{\text{SIGN}}$ is the $n \times N_A$ matrix, with the first $N_A$ principal components as columns. Next, ICA was applied to the matrix $Z_{\text{PROJ}}$, using the fastICA Matlab package. The resulting unmixing matrix $W$ was used to determine each cells weight within the assemblies:

$$V = P_{\text{SIGN}} W \tag{5}$$

where the columns of $V$ are the weight vectors of the assembly patterns.

*Unilateral and bilateral assembly detection*. Within each assembly pattern, neurons whose weight exceeded the mean weight of the population by 2 SDs was considered a member of the assembly. For each cell assembly, if the member neurons all belong to the same hemisphere (either left or right), then this cell assembly was considered as a unilateral assembly. In contrast, if a cell assembly consisted of neurons belong to both the left and right hemispheres, it was considered as a bilateral assembly. The observed ratio of unilateral assemblies was derived by the number of unilateral assemblies divided by the total number of assemblies when the animal was moving in each direction.

*Shuffling of member neurons in each cell assembly*. For the cell assemblies that were detected in each animal, each cell assembly was reconstituted with a random neuron recorded from this animal non-repetitively and the ratio of unilateral assemblies for each animal was calculated as described above. This reconstitution was repeated for 1000 times, as a result, 1000 shuffled ratios of unilateral assemblies were obtained.

*Tracking expression of assembly patterns over time*. Assembly reactivation was evaluated after the identification of cell assemblies in the exploration session, as follows:

$$R_k(t) = z(t)^T P_k z(t) \tag{6}$$

where $R_k(t)$ is the activation strength of assembly $k$ at time $t$ and $z(t)$ is the z-scored binned spike time convolved with the Gaussian kernel. $P_k$ is the projection matrix of pattern $k$ and was constructed from the outer product of its weight vector $v_k$, in which diagonal entries were set to zero. Assembly strength was then calculated as the mean strength of each assembly's pattern activations, with only peaks in the expression strength exceeding 5 included. Assemblies with the reactivation rate < 0.01/s were excluded from the statistic test.

*Ripple detection*. SWRs were only detected when animal velocity was below 2 cm/s. SWRs were detected by first band-pass filtering the local-field potential within the frequency range of 90 and 250 Hz using a Hamming window-based FIR filter and calculating the root mean square power in the ripple band. SWR events were defined as periods, where peak power was more than 5 SD above the mean, and began and terminated at the time when power was < 2 SD. In addition, two ripple events were considered as one if their peaks were closer than 50 ms. Peak frequency was calculated using the multitaper method.

*Ripple synchronization*. The temporal start points and stop points of all detected ripple events were used to calculate the temporal midpoint of each events. A vector of these timestamps was constructed for each hemisphere and with temporal midpoints of ripple events from one hemisphere used as reference, the occurrence of ripple events in the other hemisphere was detected across a 200 ms window centered on each midpoint of events from reference hemisphere with 1 ms bin size. The resulted column vector represented the probability of ripple events from the opposite hemisphere and was then smoothed with a Gaussian kernel with an SD of 4.

*Bayesian replay analyses*. Spikes of each place cell were separated into two groups according to the direction of the animal's movement. Only ripple events with a duration between 50 and 300 ms, and in which at least five place cells participated were included in the further analysis. Each ripple event was binned into non-overlapping 20 ms windows and spike activity of place cells during the ripple event was constructed within those bins. The activity of place cells in each window was decoded to generate a virtual position with the Bayesian method[37], with the smoothed FR of place cells during previous exploration:

$$\Pr(\text{pos}|\text{spikes}) = \left(\prod_{i=1}^{n} f_i(\text{pos})^{sp_i}\right) e^{-\tau \sum_{i=1}^{n} f_i(\text{pos})} \tag{7}$$

where $f_i(\text{pos})$ is the FR of the template of $i$th place cell at the position pos, $sp_i$ is the number of spikes of the $i$th place cell in the temporal bins being decoded, $\tau$ is the temporal duration of decoding bins, and $n$ is the total number of place cells. Then the posterior probabilities were normalized to 1 as follows:

$$\Pr(\text{pos}|\text{spikes}) = \frac{\Pr(\text{pos}|\text{spikes})}{\sum_{i=1}^{P_n} P_r(\text{pos}_i|\text{spikes})} \tag{8}$$

where $P_n$ is the total number of positions.

Subsequently, a preliminary sequence score ($r$) was derived as the correlation of time and position for each event and weighted by posterior probability. First, the weighted mean ($m$) was calculated as follows:

$$m(\text{pos};\Pr) = \frac{\sum_{i=1}^{M} \sum_{j=1}^{N} \Pr_{ij} \text{pos}_j}{\sum_{i=1}^{M} \sum_{j=1}^{N} \Pr_{ij}} \tag{9}$$

Next, the weighted covariance (cov) was calculated:

$$\text{cov}(\text{pos}, \text{bin};\Pr) = \frac{\sum_{i=1}^{M} \sum_{j=1}^{N} \Pr_{ij} \left(\text{pos}_j - m(\text{pos};\Pr)\right)\left(\text{bin}_i - m(\text{bin};\Pr)\right)}{\sum_{i=1}^{M} \sum_{j=1}^{N} \Pr_{ij}} \tag{10}$$

Then the weighted correlation $r(\text{pos}, \text{bin}; \Pr)$ was calculated:

$$r(\text{pos}, \text{bin};\Pr) = \frac{\text{cov}(\text{pos}, \text{bin};\Pr)}{\sqrt{\text{cov}(\text{pos}, \text{pos};\Pr)\text{cov}(\text{bin}, \text{bin};\Pr)}} \tag{11}$$

where $\text{pos}_j$ is the $j$th spatial bin, $\text{bin}_i$ is the $i$th temporal bin and $\Pr_{ij}$ is the posterior probability in that bin. $M$ is the total number of temporal bins and $N$ is the total number of spatial bins.

Next, the FR template of each place cell was circularly shuffled 1000 times. The activity of place cells in each window during each ripple event was then decoded again via the same method described above using the 1000 shuffled templates to generate 1000 shuffled weighted correlations (1000 $r(\text{null})$). For each event, a sequence score ($rZ$) was derived by taking the absolute value of the observed weighted correlation $r(\text{observed})$ as a z-score relative to its expected absolute null weighted correlation distribution $r(\text{null})$:

$$rZ = \frac{|r(\text{observed})| - \overline{|r(\text{null})|}}{\text{SD}(|r(\text{null})|)} \tag{12}$$

In addition, in order to identify the quality of replay, a Monte Carlo P-value[38] was calculated for each event as follows:

$$P = \frac{(n+1)}{(r+1)} \tag{13}$$

where $r$ is the number of shuffles (1000), $n$ is the number of shuffles with a weighted correlation greater than the observed value. As each event was decoded twice for different directions of movement, only the event with the smallest $P$-value and only if that $P$-value was < 0.05 were events included for further analysis.

Mean jump distance was derived as the average distance between peak decoded positions in adjacent temporal bins of each replay event. Replay distance was derived as the distance between the maximum and minimum of peak decoded positions in each replay event. Replay speed was derived as the replay distance divided by the time period that this replay event spent.

*PCC analyses.* The PCC of each cell[22] in each event was derived by calculating the difference between the observed sequence score of the event that the cell participated in ($rZ_e$(observed)) and the specific single-cell shuffled sequence score of the same event ($rZ_{e,c}$(cell $c$ shuffled))), normalized by the number of participating cells in this event:

$$\text{PCC}_{e,c} = \left[ rZ_e(\text{observed}) - rZ_{e,c}\left(\text{cell} \quad c \quad \text{shuffle}\right)\right] \times \text{Participants}_e \qquad (14)$$

where #Participants$_e$ is the number of place cells that fired more than one spike in event $e$. $rZ_e$(observed) was derived as described above. For each cell that participated in the replay event, only the tuning curve for that specific cell was circularly shuffled 1000 times and the average of this 1000 single-cell shuffled sequence scores is the $rZ_{e,c}\left(\text{cell} \quad c \quad \text{shuffle}\right)$. The PCC of each neuron was defined as the neuron's mean contribution across all significant replay events.

*Statistics and reproducibility.* All data were analyzed with MATLAB R2019b. Wilcoxon's rank-sum test and unbalanced two-way analysis of variance was applied to test the difference between groups. The Bonferroni test was used to conduct multiple comparisons between multiple groups for post hoc tests. Linear mixed effects models were applied to test the difference between the left and right hippocampus, where mouse identity was specified as a random factor, and the left and right were specified as fixed factors. Results are shown as mean ± SEM. All box plots in Figs. 1c, d, 2b, c, 3b, 4b, c, 5, and Supplementary Figs. 2–4 represent the distribution of data, with the median represented by the central mark, 25th and 75th percentiles represented by the bottom and top edges of the box, respectively, and the whiskers extend to the most extreme data points not considered outliers; data are considered as outliers if they are > q3 + 1.5 × (q3 − q1) or < q1 − 1.5 × (q3 − q1), where q1 and q3 are the 25th and 75th percentiles of the sample data, and the outliers are shown invisible in order to make a suitable visualization; ****$P < 0.0001$, ***$P < 0.001$, **$P < 0.01$, *$P < 0.05$. Representative example data in the following figures were selected from multiple experiments with similar results: Fig. 1b: 4 examples of 152 neurons recorded from the left CA1 and 4 examples of 121 neurons recorded from the right CA1 in CTRs, 4 examples of 162 neurons recorded from the left CA1 and 4 examples of 156 neurons recorded from the right CA1 in MUTs; Fig. 2a, c: 1 of 4 CTR animals and 1 of 5 MUT animals; Fig. 3c: 6 of 277 replay events detected from 4 CTR animals and 6 of 182 replay events detected from 5 MUT animals; Supplementary Fig. 1: 1 of 9 animals.

**Reporting summary**. Further information on research design is available in the Nature Research Reporting Summary linked to this article.

## Data availability

Source data are available at https://doi.org/10.5281/zenodo.5529405. Processed data are provided with this paper.

## Code availability

The custom MATLAB scripts used in this study are available at https://github.com/HefeiGuan/Guan-et-al-Nat-Comm-2021.

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

## Acknowledgements

We thank M. Fujisawa for daily assistance, the Advanced Manufacturing Support Team at RIKEN Center for Advanced Photonics for microdrive production, all the members of the Laboratory for Circuit and Behavioral Physiology for their support on the experiments and manuscript, and Dr. Takeru Matsuda for advice on statistical testing. This work was supported by RIKEN CBS, a Grant-in-Aid for Scientific Research from MEXT (19H05646; T.J.M.), a JRP-LEAD grant from JSPS (20181703; T.J.M.), and a Grant-in Aid for Scientific Research on Innovative Areas from MEXT (19H05233; T.J.M.).

## Author contributions

H.F.G., S.J.M., and T.J.M. designed the experiments. T.I. and T.J.M. provided supervision. H.F.G. performed the experiments and data analyses with the support of S.J.M. H.F.G. and T.J.M. wrote the manuscript with input from all authors.

## Competing interests

The authors declare no competing interests.
