## [Peer Review File · Nature Communications]

Lateralization of CA1 assemblies in the absence of CA3 inputReviewers' Comments:

Reviewer #1:

Remarks to the Author:

Using state-of-the-art in vivo electrophysiology recordings and analysis, in good part pioneered by the authors, in combination with genetically targeted silencing of afferents, this study investigates the role of CA3 to CA1 input in the bilateral coordination of CA1 spatial coding, both during behavior and periods of consolidation. The study fills a well-defined, albeit small, gap in our understanding of lateralised activity in rodent hippocampus. The main result is that bilateral CA3 to CA1 input is important for subtle tuning of cell assembly activity (shown using information theory approaches and cross-correlations).

The study design, approach and analysis are of high quality. The manuscript is written lucidly and with clear logic progression. The study's findings are clearly presented and their implication/impact appropriately discussed (i.e. not over-stated).

Overall, this is a very nice, short study that builds on a rather large body of prior work (including from the authors).

Reviewer #2:

Remarks to the Author:

Guan and colleagues use the CA3-TetX mouse to assess the impact of blocking CA3-CA1 transmission (uni- and bilaterally) on place cell activity characteristics and cross-hemisphere coordination. This genetic model has been studied extensively before (by some of the authors of the paper), thus effects on place cell activity and SWRs has already been established. What's new in this study is that the authors show bilateral CA1 coordination is affected when CA3 input to CA1 is blocked. This is an interesting result although not particularly surprising (cross-hemisphere coordination has to be mediated by CA1 commissural connections which are blocked here).

I have some concerns about the results obtained that relate to inconsistency with other previous studies and analytical confounds that may influence their main result. I explain these concerns below. It should be noted all suggested further analyses are easy to implement.

1) Firstly, if the authors have silenced CA3 how do they suggest replay and SWRs are generated?

2) Rate findings contradict those of Davoudi & Foster. They observed lower rates of CA1 cells in the presence of acute CA3 silencing. Could the authors confirm other rate metrics are also similar between their two mouse groups - e.g. max place field rate which was found to be reduced in Davoudi & Foster. If they find their effects are confirmed with other metrics could the authors comment on why they do not find the same effects as Davoudi & Foster did?

If the authors do find rate differences between mutants and controls after checking other rate measures they need to control for that in all assembly analyses

3) The authors find less cross-hemisphere coordination during HPC replay. However, the authors also find SWRs are shorter, lower in amplitude and less likely to occur in the mutants. Can they check these differences are not confounding their cross-hemisphere coordination result? e.g. if they match the duration of mutant and control SWRs do they still find mutants have more lateralised replay?

4) Also, supplementary figure 2 shows RH in mutants have longer and higher amplitude SWRs than LH. The panels in this figure suggest no difference is statistically significant though. Can the authors a) double check no L vs R mutant comparison is significant (if you come back to the boxplots in this figure to those in some of the main figures which have a significant result it's very surprising none of these supplementary results are not significant). b) Check if RH CA1 cells are more likely to participate in LH replay than LH CA1 cells? If this is not the case this also qualifies any concerns about differences in

SWR characteristics confounding the main results of the paper.

Minor comments:

Fig2a: why does mutant raster indicate higher assembly strength than controls? The authors claim assembly strength is lower in mutants but the raw data does not represent this well.

Labelling of panels is very scarce - e.g. fig3d has 5 plots but all belong to panel D. This makes it hard to follow the legend.

Reviewer #3:

Remarks to the Author:

Guan et al. examine the effects of chronic CA3 inhibition on interhemispheric coordination of neural activity in CA1. Loss of CA3 function abolishes coordination of cell assemblies across hemispheres, such that groups of neurons with coordinated activity are seen to a greater degree within hemispheres than across hemispheres compared to controls. Thus, it appears that CA3 is essential for coordinating hippocampal neural activity across hemispheres. This paper is a timely and interesting contribution to the neurobiological study of memory and is within the scope of Nature Communications. Despite this, I have several concerns:

1) This study uses only male mice without any stated justification (see Shansky, 2019). Unless justified, female mice should be included in both cohorts.

2) This study makes several comparisons of the left and right hippocampi using tetrode recordings. To do this, it must be clear that there are no systematic differences of tetrode positions across hemispheres. In other words, does a right-handed experimenter perform the implantation surgery so that the tetrodes in the right hemisphere are angled towards the midline while tetrodes in the left hemisphere are away from the midline? At a minimum, histology should be presented of the bilateral hippocampus with recording sites and tetrode tracks.

3) The authors need to justify the use of Wilcoxon rank sum tests used here. There is evidence that the data presented in this manuscript are highly skewed. For instance, the median values depicted for assembly strength in figures S3a and S3b are very different from the means reported in the text. Wilcoxon rank sum tests are not appropriate for highly skewed data (McElduff et al., 2010). Further, the Wilcoxon rank sum test assumes independence of data points and these data are not independent (multiple data points from individual mice). Linear mixed effects models are appropriate to capture interindividual variability and can be easily performed in R.

4) The discussion should be expanded. The authors cite several in vitro studies demonstrating lateralization of hippocampal circuits, but do not discuss any in vivo studies, such as Song et al. (2020, Nat Comm), Benito et al. (2016, eLife) and Villalobos et al. (2017, PLoS One). Discussing the present data in the context of what was learned in these studies would enhance our understanding of how the hippocampus is organized bilaterally. Further, I would be interested in a discussion of how loss of CA1 bilateral coordination could be related to the memory deficits seen following loss of CA3 function. As the authors have established in previous work, not all hippocampus-dependent memories require CA3- why might these types of memories not require interhemispheric coordination?

5) It is not clear what the difference is between the results reported in the middle and bottom paragraphs. It appears that the middle paragraph may be all recorded cells pooled across hemispheres and compared between controls and mutants while the bottom paragraph contains the same analysis

but with comparisons across hemispheres in addition to control/mutant. If so, this seems redundant and the analysis of cells pooled across hemisphere can be removed, unless a reason can be given in the text.

6) Figure 2: Please increase the font size in 2a and make the raster plot tick marks darker and/or more prominent. The yellow in particular is hard to see.

7) Manuscript should be reviewed for typos/grammar.

References

- Benito et al. (2016), eLife, doi: [10.7554/eLife.16658](https://doi.org/10.7554/eLife.16658)
McElduff et al. (2010), Adv Physiol Educ, doi: [10.1152/advan.00017.2010](https://doi.org/10.1152/advan.00017.2010)
Shanksy (2019), Science, doi: [10.1126/science.aaw7570](https://doi.org/10.1126/science.aaw7570)
Song et al. (2020), Nat Comm, doi: [10.1038/s41467-020-16698-4](https://doi.org/10.1038/s41467-020-16698-4)
Villalobos et al. (2017), PLoS One, doi: [10.1371/journal.pone.0171304](https://doi.org/10.1371/journal.pone.0171304)

Reviewed by Jake Jordan

Draft point by point response for NCOMMS-21-10509A-Z

Reviewer comments in **black**, responses in **blue**, notes to editorial team in **red**

Reviewer #1 (Remarks to the Author):

Using state-of-the-art in vivo electrophysiology recordings and analysis, in good part pioneered by the authors, in combination with genetically targeted silencing of afferents, this study investigates the role of CA3 to CA1 input in the bilateral coordination of CA1 spatial coding, both during behavior and periods of consolidation. The study fills a well-defined, albeit small, gap in our understanding of lateralised activity in rodent hippocampus. The main result is that bilateral CA3 to CA1 input is important for subtle tuning of cell assembly activity (shown using information theory approaches and cross-correlations).

The study design, approach and analysis are of high quality. The manuscript is written lucidly and with clear logic progression. The study's findings are clearly presented and their implication/impact appropriately discussed (i.e. not over-stated).

Overall, this is a very nice, short study that builds on a rather large body of prior work (including from the authors).

We thank the reviewer for the supportive comments.

The only point we would like to highlight, for the benefit of all the referees, is that while we agree the current manuscript does build on our past work, we believe it adds two important advances.

First, in terms of the circuitry underlying replay, hitherto the current work we had been unable to look at the reactivation of place cell trajectories during SWRs in these animals due to the technical challenges of recording a sufficient number of neurons simultaneously in the mouse. Here, our high-density bilateral tetrode recording permitted this analysis for the first time and we find, contrary to expectation, that even with CA3 transmission completely silenced, replay does persist, albeit with some significant alterations and, importantly, in a more lateralized manner. As suggested by Reviewer #2 we now address this finding directly in the revised discussion section and speculate on possible mechanisms.

Second, related to the circuits contributing to the integration of spatial information across hemispheres in the hippocampus; i.e lateralization; while there are several in vivo physiological studies that mention hippocampal lateralization, either in passing (Carr et al, Neuron 2012; Pfeiffer & Foster, Nature 2015) or more directly (Benito et al, eLife, 2016; Villalobos et al, PLoS One, 2017), a careful and complete examination of the physiological properties at both the single cell and population level across the hemispheres in freely behaving rodents is absent. Further, no previous work has employed interventional approaches to address the circuits responsible for the integration of spatial information across the left and right hippocampus; here we do both.

Reviewer #2 (Remarks to the Author):

Guan and colleagues use the CA3-TetX mouse to assess the impact of blocking CA3-CA1 transmission (uni- and bilaterally) on place cell activity characteristics and cross-hemisphere coordination. This genetic model has been studied extensively before (by some of the authors of the paper), thus effects on place cell activity and SWRs has already been established. What's new in this study is that the authors show bilateral CA1 coordination is affected when CA3 input to CA1 is blocked. This is an interesting result although not particularly surprising (cross-hemisphere coordination has to be mediated by CA1 commissural connections which are blocked here).

I have some concerns about the results obtained that relate to inconsistency with other previous studies and analytical confounds that may influence their main result. I explain these concerns below. It should be noted all suggested further analyses are easy to implement.

We thank the reviewer for their interest and constructive suggestions, our improvements to the manuscript based on your comments are detailed below.

Additionally, we would like to offer a point of clarification on the CA3-TetX mice and the circuitry silenced in the model. This triple transgenic mouse line allows inducible and complete silencing of all CA3 pyramidal cell transmission in the adult animal (Nakashiba et al 2008), including, as we note in the paper, CA3 to CA1 ipsi and contralateral transmission, all CA3 to CA3 recurrent transmission, CA3 to CA2 transmission (ipsi & contra), and any subcortical outputs. As the reviewer references *CA1 commissural connections* above, we would like to say that these projections, to the limited extent they exist in the rodent, would be intact in this mouse.

1) Firstly, if the authors have silenced CA3 how do they suggest replay and SWRs are generated?

This is an important point and we apologize for not directly addressing this in the initial version of the manuscript. This omission was due to the fact the paper was written in a Brief Communications format and space was limited. In the revised version we have added a paragraph that directly speaks to our ideas on how replay and SWRs remain present in the absence of CA3 transmission. In short, while we believe that in an intact brain CA3 is indeed the primary driver of these events, there is mounting evidence that SWRs can be triggered by other noncanonical mechanisms, including CA2, the subiculum and perhaps even intrinsically in CA1. These data are highlighted below:

- Recent in vivo evidence has established that activity in CA2 can, and does, contribute both to SWR generation and reactivation of assemblies, even when CA3 is intact (Oliva et al 2016, Oliva et al 2020). Single-unit recordings from all CA subregions in vivo demonstrate that prior to the characteristic bursting observed in CA3, which classically signaled the onset of SWRs, the CA2 pyramidal cell population can display a biphasic activity pattern preceding its neighbors. Initially, deep CA2 cells (termed ramping cells) show a sustained enhancement in their firing rates several hundred milliseconds prior to SPW-Rs, before the superficial CA2 cells (termed phasic cells) display the large discharge increases characteristic of SWR-associated neuronal

firing. This activity, with brief temporal delays, is then successively evident in areas CA3a, CA3b, CA3c, and finally CA1. Moreover, in the 2020 Oliva et al study the authors demonstrated structured reactivation of socially active neurons during these CA2 SWR events, as well as evidence that manipulation of these events could impact memory.

- Next, a very recent study (Imbrosci et al, 2021 Cell Reports) provided both in vitro and in vivo evidence that the subiculum can serve as a secondary SWR generator, events that propagate backwards into the hippocampus. Although that study did not examine replay, it raises the possibility that SWRs seen in the absence of CA3 transmission may also have a subicular origin.
- Finally, there is evidence from in vitro studies that CA1 can generate SWRs independently, when all other hippocampal subfields are dissected away (Maier et al., 2003).

With our current data we cannot distinguish between these possibilities. However, given that CA3 acts primarily to inhibit CA2 in vivo (Boehringer et al 2017) and even under normal conditions CA2 is responsible for generating a proportion of normal SWRs in CA1, our belief is that CA2 most likely acts to initiate SWRs in the absence of CA3 output. This is now detailed in the discussion.

2) Rate findings contradict those of Davoudi & Foster. They observed lower rates of CA1 cells in the presence of acute CA3 silencing. Could the authors confirm other rate metrics are also similar between their two mouse groups - e.g. max place field rate which was found to be reduced in Davoudi & Foster. If they find their effects are confirmed with other metrics could the authors comment on why they do not find the same effects as Davoudi & Foster did?

If the authors do find rate differences between mutants and controls after checking other rate measures they need to control for that in all assembly analyses

We agree that a careful consideration of the similarities and distinctions between the data in the Davoudi & Foster paper and data generated from recordings in the CA3-TetX mice, both in the current paper, as well as in previous studies, is important for a full understanding of the mechanisms of SWRs and replay. First, in the Davoudi & Foster study the authors employed acute optogenetic silencing of CA3 axons in the stratum radiatum of CA1 and examined the impact on place fields, as well as SWRs and replay. They observed that during the "light ON" condition the average peak rate of CA1 pyramidal cells decreased dramatically, however this is not the case in our model, as noted in the 2008 Nakashiba et al Science paper which first characterized this transgenic line, as well as in our current recordings.

Moreover, Davoudi & Foster observed a significant decrease in SWR occurrence, as well as in the concurrent multiunit spiking activity of CA1 pyramidal cells during optogenetic silencing, which they readily admit are serious confounds to their replay analysis. In contrast, in our transgenic model we employ chronic silencing and do not observe either of these confounding changes. Further, as Davoudi & Foster note, their approach is not only temporally acute, it is also spatially limited, only impacting a fraction of the recorded neurons (see their Fig. 1j) and not targeting the CA3 axons impinging on the basal CA1 dendrites in str. oriens, thus complicating the interpretation of their findings. Again in contrast, our transgenic model permits complete silencing of transmission across all CA3 pyramidal cells (see Nakashiba et al, Science 2008), providing a more solid, interpretable and reproducible model to study these phenomena and answer the question of exactly what CA3 input contributes. We believe that the complementary nature of acute and chronic approaches has established that both are critical

for the field to move forward and we have added these points in the discussion section of the revised manuscript.

3) The authors find less cross-hemisphere coordination during HPC replay. However, the authors also find SWRs are shorter, lower in amplitude and less likely to occur in the mutants. Can they check these differences are not confounding their cross-hemisphere coordination result? e.g. if they match the duration of mutant and control SWRs do they still find mutants have more lateralised replay?

We apologize if our data were presented in an unclear manner in the original version of the manuscript. In fact, we find no differences in ripple duration, amplitude or occurrence in the CA3-TeTX mice, only a decrease in intrinsic frequency, which has been previously reported in this model (Nakashiba et al, 2009; Middleton & McHugh, 2016). In the current manuscript this was reported on page 4, where we wrote, *“Consistent with the previous results in the CA3-TeTX mice^{5,7}, we observed a decrease in the intrinsic SWRs frequency in the mutant mice, while the amplitude, occurrence and duration were similar to controls (Frequency: CTR, 132.9 ± 5.54 Hz; MUT, 108.3 ± 2.01 Hz; Amplitude: CTR, 0.25 ± 0.020 mV; MUT, 0.22 ± 0.026 mV; Duration: CTR, 88.79 ± 5.569 ms; MUT, 81.74 ± 3.152 ms; Occurrence: CTR, 0.21 ± 0.018 /s; MUT, 0.17 ± 0.011 /s; **Supplementary Fig. 2a).**”*

4) Also, supplementary figure 2 shows RH in mutants have longer and higher amplitude SWRs than LH. The panels in this figure suggest no difference is statistically significant though. Can the authors a) double check no L vs R mutant comparison is significant (if you compare the boxplots in this figure to those in some of the main figures which have a significant result it's very surprising none of these supplementary results are not significant). b) Check if RH CA1 cells are more likely to participate in LH replay than LH CA1 cells? If this is not the case this also qualifies any concerns about differences in SWR characteristics confounding the main results of the paper.

a) We **will** recheck the statistical tests on the data in supplementary figure 2 and confirm there is no significant differences between the genotypes. In these plots we used each mouse as a single data point (N=4 for control, N=5 for CA3-TeTX), which explains the large variation in some of the measures. While we believe this is the correct way to group the data, we **will** recalculate the data and statistics using each ripple as an independent sample and confirm no hemispheric differences.

b) We **will** examine this and confirm that in control animals there is no difference in participation of RH and LH pyramidal cells in LH ripples, while in the mutant mice RH cells are less likely to fire during those events.

Minor comments:

Fig2a: why does mutant raster indicate higher assembly strength than controls? The authors claim assembly strength is lower in mutants but the raw data does not represent this well.

We thank the reviewer for raising this point. As shown in Figure 2b, there is actually a significant decrease in assembly strength in the mutants. Obviously, the group data represents many runs on the linear track from multiple animals, which vary across subjects and runs. The example we chose may not reflect the group data, thus we **will** search for a more representative example to replace it with.

Labelling of panels is very scarce - e.g. fig3d has 5 plots but all belong to panel D. This makes it hard to follow the legend.

We apologize; we have reformatted the figures in the revised manuscript to try to improve readability.

Reviewer #3 (Remarks to the Author):

Guan et al. examine the effects of chronic CA3 inhibition on interhemispheric coordination of neural activity in CA1. Loss of CA3 function abolishes coordination of cell assemblies across hemispheres, such that groups of neurons with coordinated activity are seen to a greater degree within hemispheres than across hemispheres compared to controls. Thus, it appears that CA3 is essential for coordinating hippocampal neural activity across hemispheres. This paper is a timely and interesting contribution to the neurobiological study of memory and is within the scope of Nature Communications. Despite this, I have several concerns:

We thank Dr. Jordan for his support and excellent suggestions, we believe the manuscript has been improved as a result.

1) This study uses only male mice without any stated justification (see Shansky, 2019). Unless justified, female mice should be included in both cohorts.

Thank you for raising this important point. Indeed, we have previously discussed the importance of the commentary from Dr. Shansky in our group and fully agree that the consideration of sex as a biological variable is crucial for the future of neuroscience. Further, we acknowledge that the use of only male mice in this study is a limiting factor and this will be explicitly stated in the revised text. The justification, if that word is warranted here, for the use of only male mice in the current work is that this is a triple transgenic line in which the inducible expression of TeTX is triggered by the removal of doxycycline from the animals' diet. This line of mice was engineered roughly 15 years ago and the careful analysis of the drug levels, the kinetics of TeTX expression following drug withdrawal and the resulting silencing of CA3 transmission, verified in vitro and in vivo, were only performed in male mice. While it is likely the parameters would be similar in females, these data must be generated in order to ensure accuracy in our conclusions; this is something we will address in the future. However, due to the restrictions on breeding and housing of animals over the last 13 months due to the pandemic it would be extremely challenging for us to complete that work and add physiological recordings from female mice in a reasonable amount of time in the current study.

2) This study makes several comparisons of the left and right hippocampi using tetrode recordings. To do this, it must be clear that there are no systematic differences of tetrode positions across hemispheres. In other words, does a right-handed experimenter perform the implantation surgery so that the tetrodes in the right hemisphere are angled towards the midline while tetrodes in the left hemisphere are away from the midline? At a minimum, histology should be presented of the bilateral hippocampus with recording sites and tetrode tracks.

Thank you for the suggestion. We have now added histology demonstrating no systematic differences in position of tetrodes between the hemispheres. Moreover, all our data is collected using individually adjustable tetrodes which are repositioned daily based on physiological markers such as SWR amplitude and multiunit spike activity to ensure proper positioning within the CA1 pyramidal cell layer.

3) The authors need to justify the use of Wilcoxon rank sum tests used here. There is evidence that the data presented in this manuscript are highly skewed. For instance, the median values depicted for assembly strength in figures S3a and S3b are very different from the means reported in the text. Wilcoxon rank sum tests are not appropriate for highly skewed data (McElduff et al., 2010). Further, the Wilcoxon rank sum test assumes independence of data points and these data are not independent (multiple data points from individual mice). Linear mixed effects models are appropriate to capture interindividual variability and can be easily performed in R.

We thank the reviewer for this suggestion. We will review our choice of statistical testing and will employ LMMs when appropriate.

4) The discussion should be expanded. The authors cite several in vitro studies demonstrating lateralization of hippocampal circuits, but do not discuss any in vivo studies, such as Song et al. (2020, Nat Comm), Benito et al. (2016, eLife) and Villalobos et al. (2017, PLoS One). Discussing the present data in the context of what was learned in these studies would enhance our understanding of how the hippocampus is organized bilaterally. Further, I would be interested in a discussion of how loss of CA1 bilateral coordination could be related to the memory deficits seen following loss of CA3 function. As the authors have established in previous work, not all hippocampus-dependent memories require CA3- why might these types of memories not require interhemispheric coordination?

We fully agree with the reviewer's comment. We originally submitted this work for consideration as a Brief Communication hence the rather short discussion section. In hindsight, we would have been much better off expanding our discussion and have addressed all the above points in the revised version. We will include:

- A discussion of our findings in light of the three in vivo papers cited, highlighting the novel contributions of our current work.
- A discussion of the possible behavioral impact related to the increased lateralization of the representations we observe, as well as why certain behaviors remain intact despite CA3 silencing (such as the Morris Water Maze, see Nakashiba et al 2008).

5) It is not clear what the difference is between the results reported in the middle and bottom paragraphs. It appears that the middle paragraph may be all recorded cells pooled across hemispheres and compared between controls and mutants while the bottom paragraph contains the same analysis but with comparisons across hemispheres in addition to control/mutant. If so, this seems redundant and the analysis of cells pooled across hemisphere can be removed, unless a reason can be given in the text.

We apologize for the confusion. Due to the fact that replay has not yet been reported in these mice (see our comment to reviewer #1 above), we thought it was crucial to first characterize replay using all cells. Thus, the analysis in Figure 3d used pooled cells from both left and right hemispheres, establishing replay events are present following complete chronic silencing of CA3, although the quality of the events is worse. Given our interest in lateralization, next in Figure 3f we used cells from a single hemisphere, either left CA1 or right CA1, and the cell spikes during ripple from same or opposite hemisphere were used to decode replay. The results in figure 3f suggested silencing CA3 weakens coordinated memory replay across bilateral CA1.

6) Figure 2: Please increase the font size in 2a and make the raster plot tick marks darker and/or more prominent. The yellow in particular is hard to see.

Thank you for the suggestion, this has been corrected.

7) Manuscript should be reviewed for typos/grammar.

Thank you for the suggestion, we have carefully reviewed the revised version for errors in grammar.

References

- Benito et al. (2016), eLife, doi: 10.7554/eLife.16658
McElduff et al. (2010), Adv Physiol Educ, doi: 10.1152/advan.00017.2010
Shanksy (2019), Science, doi: 10.1126/science.aaw7570
Song et al. (2020), Nat Comm, doi: 10.1038/s41467-020-16698-4
Villalobos et al. (2017), PLoS One, doi: 10.1371/journal.pone.0171304

Reviewed by Jake Jordan

Reviewers' Comments:

Reviewer #2:

Remarks to the Author:

I am pleased to see the authors have added a discussion regarding the source of replay in their model as well as noting the possibility that due to the chronic nature of their CA3 disruption this could also cause adaptations in the circuit (albeit making the interpretation of the replay source more difficult). I am satisfied with the authors' discussion although it still leaves open the question of where the replay is coming from in their model and whether these extra-ca3 replay initiators would be 'normally' recruited if CA3 is absent or whether they are being recruited due to some plasticity associated with chronic inactivation of CA3. However, the answer to these questions is beyond the scope of the authors' paper, which I accept.

The authors have addressed my concerns regarding place cell rate differences and differences in replay characteristics. However, one (small) concern arose as I read the revised manuscript: the authors show the intrinsic frequency of mutant ripples is 109Hz. This seems very low - indeed it borders with the higher end of high gamma range. Perhaps there are specie differences when it comes to ripple frequency, but in rats the peak frequency is usually well above 120Hz (often close to 200Hz). To ensure the authors are not just looking at fast gamma in their analysis, could they check their results hold with a less liberal ripple filter (currently their filter is 90-250Hz, 90Hz is right in the middle of fast gamma frequency range so a higher lower bound for their ripple filter is required).

Reviewer #3:

Remarks to the Author:

The authors' response was very clear and thoroughly addressed my concerns. Congratulations to them on an interesting paper.

Point by point response for NCOMMS-21-10509B

Reviewer comments in **black**, responses in **blue**.

Reviewer #2 (Remarks to the Author):

I am pleased to see the authors have added a discussion regarding the source of replay in their model as well as noting the possibility that due to the chronic nature of their CA3 disruption this could also cause adaptations in the circuit (albeit making the interpretation of the replay source more difficult). I am satisfied with the authors' discussion although it still leaves open the question of where the replay is coming from in their model and whether these extra-ca3 replay initiators would be 'normally' recruited if CA3 is absent or whether they are being recruited due to some plasticity associated with chronic inactivation of CA3. However, the answer to these questions is beyond the scope of the authors' paper, which I accept.

The authors have addressed my concerns regarding place cell rate differences and differences in replay characteristics. However, one (small) concern arose as I read the revised manuscript: the authors show the intrinsic frequency of mutant ripples is 109Hz. This seems very low - indeed it borders with the higher end of high gamma range. Perhaps there are specie differences when it comes to ripple frequency, but in rats the peak frequency is usually well above 120Hz (often close to 200Hz). To ensure the authors are not just looking at fast gamma in their analysis, could they check their results hold with a less liberal ripple filter (currently their filter is 90-250Hz, 90Hz is right in the middle of fast gamma frequency range so a higher lower bound for their ripple filter is required).

We thank the reviewer for the supportive comments and the opportunity to clarify this point.

First, we would like to note that the reviewer is correct, there is indeed a species difference in ripple frequency between mice and rats, with ripples in wild-type/control mice averaging in the range of 130~150Hz compared to the roughly 170~190 Hz seen in rats (see Mou et al, 2018; <https://doi.org/10.3389/fncel.2018.00332> for a detailed side by side comparison).

Next, two previous studies using the same mouse line employed in this current work (Nakashiba et al, 2009; Middleton and McHugh 2016) have reported similar significant decreases in intrinsic ripple frequency; we have included Figure 2B, from Nakashiba, Buhl, McHugh et al. 2009 (see Figure R1) as a reference for the distribution of power across the ripple band in the CA3-TeTX mice.

Figure R1: from Figure 2B, Nakashiba et al, 2009

We do realize that at the heart of the reviewer's comment is the concern that the relative low cutoff used for our ripple detection could cause contamination by fast gamma oscillations; below we will review several additional points we believe completely alleviate this possibility.

First, in the Middleton & McHugh 2016 study (see their Supplementary Figure 1b) the authors recorded across the period of transgene induction and demonstrated that the peak ripple frequency of the mutant mice decreased gradually following removal of doxycycline from the animals' diet, but remained fixed in the controls. These data suggest that the ripple detection parameters, which were consistent with the filters used here, are capable of detecting this change and differentiating it from other oscillatory activity, such as fast gamma.

Further, as also noted in our manuscript on p. 5, despite the reduction in intrinsic ripple frequency we observed, the duration of the ripples in mutants remain similar to controls, further suggesting these are indeed ripple events (see also Supplementary Figure 1b in Middleton & McHugh, 2016).

Next, in the 2016 Middleton & McHugh study the authors examined gamma oscillations present during movement, in both the slow and fast bands. They reported that power in fast gamma band is very low when the animals were stationary (velocity < 2cm/s) in both control and mutant mice (Figure R2 from Supplementary Figure 3e, Middleton and McHugh 2016; pasted below for your reference).

Figure R2: Supplementary Figure 3e, Middleton and McHugh 2016

As a velocity filter (< 2cm/s) is included in our ripple detection script, we believe it is unlikely that fast gamma events were included in our analysis. Although this threshold was in the code we used for analyzing our data in the paper, we apologized that this detail was omitted in the methods, it has now been added to the revised manuscript as highlighted below:

SWRs were only detected when animal velocity was below 2cm/s.

Nonetheless, to assure the reviewer that the changes in ripple frequency we observed following CA3 silencing is not due to the filter settings, we have redetected ripples with a narrower frequency range: 110 – 250 Hz. First, we asked what percentage of ripples remained compared to those detected with the previous range (90 - 250 Hz). Consistent with the result reported in Nakashiba et al. 2009 (Figure R1 above), when we increase the lower bound of ripple filter band from 90 Hz to 110 Hz, only ~12% of ripples were filtered out in controls, while ~28% of ripples were filtered out in mutants (Figure R3 below)

Figure R3: Ratio of ripples remaining when low-pass filter band increased from 90 to 110Hz

Again, this supports the conclusion that the ripples we detected and used in our analysis with the 90 – 250 Hz filter were indeed “genuine ripples” and did not include fast gamma. Further, we have also recalculated the cross-correlation of ripples across hemispheres with the events detected using the new ripple frequency range (110-250 Hz) (Figure R4 below) and found a virtually identical results to what we saw with the previous filter settings (Fig.3a of our manuscript). Again, suggesting the wider filter range does not affect the main results of our manuscript.

Figure R4: Cross-correlation of ripples across hemispheres with the new ripple frequency range (110-250 Hz).

In closing, we agree that the field lacks a clear standard filter setting for SWRs detection and there is variability between laboratories and published reports. The filter that we have used dates back to the Nakashiba, Buhl, McHugh et al., (Neuron, 2009) study and was explicitly designed to have the parametric space to observe both increases and decreases in average frequency. This is crucial when a genetic or behavioral intervention can impact the average frequency of the oscillation (for example see Tomar et al, 2021 which demonstrates that ripples during stress also have a significantly lower intrinsic frequency). Moreover, it is important to note that these filter settings do not provide a hard cut off, but instead lead to a substantial dampening of the frequencies in about a 10 Hz window above the low pass cut off frequency; again making fast gamma contamination an unlikely event. Finally, and perhaps most importantly, given that the ripples detected in the CA3-TetX mice do not differ from controls in all other

criteria aside from intrinsic frequency (frequency of occurrence, duration, CA1 pyramidal cell phase preference, velocity thresholding, concurrent multiunit spiking activity (see Nakashiba et al, 2009; Middleton & McHugh, 2016; and Guan et al, current submission for details), we are confident that these are indeed ripple events and distinct from fast gamma.

References

Middleton, S. J. & McHugh, T. J. Silencing CA3 disrupts temporal coding in the CA1 ensemble. *Nat Neurosci* 19, 945-951, doi:10.1038/nn.4311 (2016).

Mou X., Cheng J., Yu, Y.S.W., Kee, S.E., & Ji, D. Comparing Mouse and Rat Hippocampal Place Cell Activities and Firing Sequences in the Same Environments. *Front. Cell. Neurosci.*, 21 September 2018, <https://doi.org/10.3389/fncel.2018.00332>.

Nakashiba, T., Buhl, D. L., McHugh, T. J. & Tonegawa, S. Hippocampal CA3 output is crucial for ripple-associated reactivation and consolidation of memory. *Neuron* 62, 781-787, doi:10.1016/j.neuron.2009.05.013 (2009).

Tomar, A., Polygalov, D., Chattarji, S., & McHugh, T.J. Stress enhances hippocampal neuronal synchrony and alters ripple-spike interaction. *Neurobiol. Stress*, 13 April 2021, doi: 10.1016/j.ynstr.2021.100327.

Reviewers' Comments:

Reviewer #2:

Remarks to the Author:

I am pleased with the author's response to my comments. I appreciate the author's taking the time to thoroughly and completely address the concern I raised. I find their points to be convincing and well grounded. I have no further concerns about this manuscript.

Point by point response for NCOMMS-21-10509C

Reviewer comments in **black**, responses in **blue**.

Reviewer #2 (Remarks to the Author):

I am pleased with the author's response to my comments. I appreciate the author's taking the time to thoroughly and completely address the concern I raised. I find their points to be convincing and well grounded. I have no further concerns about this manuscript.

We thank the reviewer for the supportive comments.